# Exploring Barriers to Participation in Pediatric Rehabilitation: Voices of Children and Young People with Disabilities, Parents, and Professionals

**DOI:** 10.3390/ijerph181910119

**Published:** 2021-09-26

**Authors:** Britta Teleman, Elin Vinblad, Petra Svedberg, Jens M. Nygren, Ingrid Larsson

**Affiliations:** 1School of Health and Welfare, Halmstad University, SE-30118 Halmstad, Sweden; petra.svedberg@hh.se (P.S.); jens.nygren@hh.se (J.M.N.); ingrid.larsson@hh.se (I.L.); 2Child and Young Rehabilitation Services Kristianstad, Region Skåne, SE-29133 Kristianstad, Sweden; elinvinblad@gmail.com

**Keywords:** children, disabilities, barriers, participation, pediatric rehabilitation

## Abstract

In order to develop suitable support for participation in pediatric rehabilitation, it is important to understand what barriers need to be bridged from the perspectives of both children and adults. The aim of this study was to explore barriers to participation in pediatric rehabilitation services, according to children and young people with disabilities, parents to children with disabilities, and professionals. Data was collected in individual interviews (*n* = 48) and focus groups (*n* = 8), which were analyzed with qualitative content analysis to extract barriers to participation. Identified barriers include three categories: (1) insufficient access (controlling adults, adult-centered healthcare); (2) insufficient trust (low level of trust in adults, low level of trust in children, low self-confidence in children); and (3) insufficient involvement (norms of non-participation, low level of commitment in children). The participant groups had divergent conceptions of where and how barriers originate, and for what situations child participation is appropriate. Adult-centered healthcare and parental presence were described as barriers by all participant groups. Understanding differences in the perceptions of barriers and their origins is crucial when striving to change norms of non-participation. The findings can inform the development of new support tools and participatory formats in pediatric rehabilitation.

## 1. Introduction

The term ‘participation’ has been widely used to describe patient involvement in healthcare, although with varying definitions and levels [1]. Legislation and conventions focus on the individual patient’s right to take part in decision-making regarding their treatment [2,3], while social models hold a broader view, including the individual’s whole life situation [4]. The focus of research on participation for people with disabilities has shifted; while early research focused on physical access, recent studies emphasize the patients’ experience of involvement both in their care [5,6,7] and in daily life [8,9,10]. Most research on participation for this target group has, however, focused on home, school, and community contexts [9,11,12]. Where participation for children in healthcare is concerned the ambitions are generally high, but research has identified several obstacles for participation, many relating to dyadic situations where professionals and caregivers fail to involve the child [13,14,15,16,17,18].

While all children in healthcare may encounter obstacles for participation, children with disabilities are in even more disadvantaged positions [19]. Although the widely recognized UN Convention on the Rights of the Child (UNCRC) states that all children have the right to express their opinions in matters that affect them [3], it remains unclear what the term ‘participation’ in pediatric rehabilitation services means and how it should be translated into practice. The concept of child participation has proven problematic for professionals in pediatric rehabilitation services to grasp, both in terms of what characterizes participation for children with disabilities, and what could be considered to be a suitable level of participation for an individual child with her/his unique disability and context [5,10,20]. Professionals and parents tend to identify children’s severe physical limitations, lack of energy, and low motivation as contributing factors when asked to explain what hinders participation for a child with a given disability [21]. They thus identify the child and her/his disability as the cause for low participation. There is a risk connected to such pathogenic conceptions of disabilities as barriers, in that these barriers might be perceived as static [22]. On the contrary, research shows that the social environment is the most important factor influencing participation for children with disabilities [23,24]. 

Critical perspectives such as a social model of disability redirects the focus towards social environments. From this perspective, barriers are not an effect of an individual’s disability but by societal exclusion or contextual norms. This model highlights the importance of empowering support. Focusing on support and resources is also in line with salutogenic health promotion perspectives that have been maintained as increasing motivation for participation [25]. However, even if social contexts are researched, children and adults are known to have different experiences of and perspectives on participation [9,10,26,27]. Parents and professionals provide an adult perspective on child participation, and their views are essential to create knowledge that can support transformation towards more inclusive pediatric care [18]. Research is, however, scarce where including the views of children themselves and of multiple stakeholders are concerned [8]. Gaining such knowledge is crucial when developing participatory formats for pediatric rehabilitation, as well as for participatory research and design processes connected to such developments. The aim of this study was thus to investigate barriers to participation as identified by children and young people with disabilities, their parents, and relevant professionals in order to understand which barriers need to be bridged and how. 

## 2. Materials and Methods

### 2.1. Study Design 

This study had an explorative, qualitative design. Data were collected in individual interviews and focus groups, which were analyzed with qualitative content analysis [28,29].

### 2.2. Participants

Participants were recruited from pediatric rehabilitation services in southern Sweden. A purposeful sample of 54 participants were included: 20 children with disabilities, 8 young people with disabilities, 17 parents (this term includes all legal caregivers, e.g., foster parent, adoptive parent) to children with disabilities, and 9 healthcare professionals from different pediatric rehabilitation services (Table 1). Inclusion criteria for the children were: aged 6–17, with established contact with the pediatric rehabilitation services, and with the ability to participate in an adapted interview setting with their chosen mode of communication when answering questions. For the young people with disabilities, inclusion criteria were: aged 18–30 years, and with previously established contact with pediatric rehabilitation services. The inclusion criterion for parents was to have a child currently in contact with a pediatric rehabilitation service. For the healthcare professionals (hereafter referred to as professionals), the inclusion criteria were to have been employed at a pediatric rehabilitation service for at least one year. Furthermore, professionals were chosen to represent each profession in the multidisciplinary team. 

### 2.3. Data Collection

Data collection took place in 2017 and 2018 through individual interviews (*n* = 48) and focus groups interviews (*n* = 8). All interviews started with the researcher clarifying the aim of the study. An open interview guide was used, with initial questions focusing on participation and on what hinders participation for children within pediatric rehabilitation. In order to encourage the participants to probe more deeply into a subject, they were asked to explain their answers, by using questions such as “how do you mean?” or “what are you thinking of when you say…”. All of the interviews were undertaken by a researcher without a pre-existing or ongoing relationship with the participants, and were digitally recorded and transcribed verbatim.

The individual interviews with the children with disabilities (20–60 min, mean 25 min) were carried out by one researcher (EV), a speech and language therapist working in pediatric rehabilitation, and knowledgeable in supporting children with expressive language difficulties. The interviews were conducted at a time and place decided by the children. The children could choose to be accompanied by an adult, although only one child chose to have their parent present. 

Two focus groups with young people with disabilities were held (44 and 75 min) by a researcher (PS) and an occupational therapist (ML, EO). One group had two participants and the other had six participants. The two occupational therapists then held individual interviews (30 and 40 min) with two young people who were considered able to further contribute with essential data [30]. The participants had full access to a range of augmentative tools throughout the interviews.

The individual interviews with the parents (32–103 min, mean 59 min) were carried out by a researcher (IL) via phone or video conferencing tools (Skype or FaceTime). 

The professionals were also individually interviewed (31–59 min, mean 44 min) by phone or video conferencing tools. The interviews with the professionals were performed by one of the researchers (J.M.N., P.S.). 

### 2.4. Data Analysis

Data analysis was conducted using qualitative content analysis on both manifest and latent levels [28,29]. To ensure that the childrens’ opinions were not outweighed by the adults’ often lengthier expressions, the interviews were grouped where one group consisted of the interviews with children and young people and the other of the parents’ and professionals’ interviews. Initially, all interviews were read through several times to gain familiarity, after which relevant text was identified and marked as meaning units. Subsequently, the meaning units were abstracted, coded, and mapped into seven subcategories and three categories for each group respectively. The two groups were then compared to find similarities and differences between them. Finally, using all material, an overall analysis was iterated until consensus was reached. Two researchers (EV and IL) conducted the initial manifest analyses of all interviews and a third researcher (BT) then independently verified and refined the manifest analysis. Thereafter the whole research group reviewed and discussed the manifest level and interpreted the analysis to a latent level in an overall theme comprising the three categories.

### 2.5. Ethical Considerations

The Regional Ethical Review Board at Lund University, Sweden approved the study (No: 2017/707). The study conforms to ethical principles as set out by both national guidelines [31] as well as the ethical principles for research on human beings by the World Medical Association’s declaration of Helsinki [32]. Prior to inclusion in the study, written and oral information on the study and the voluntary nature of participation were given to the children, young people, parents, and professionals. Information to the children also included that participation or withdrawal would not affect their future healthcare in pediatric rehabilitation services. The participating children chose whether to consent only orally or in writing as well, complemented by their parents’ written consent. Written consent was also collected from the other participant groups: young people, parents, and professionals. All of the participants were offered the possibility to discuss any emotions or thoughts that may have arisen during the interviews. 

## 3. Results

The results contain identified barriers to participation for children within the pediatric rehabilitation services and include three categories: ‘*Insufficient access’*, ‘*Insufficient trust*’, and *‘Insufficient involvement*’, which together resulted in the overall theme ‘*Insufficient prerequisites for child participation*’ (Table 2). The categories are described as insufficiencies in relation to the child’s participation. When participants experienced insufficient access, trust, and involvement towards each other or the rehabilitation services, it led to low expectations of child participation in all parts. All categories were found in all participant groups, but each group differed in how frequently the subcategories were mentioned (Table 3). The results thus reveal a difference in perspectives as to what contributes to low participation.

### 3.1. Insufficient Access

The category ‘Insufficient access’ relates to the participants’ experiences of lacking access to collaboration between the child and the professional, due to ‘controlling adults’ or an ‘adult-centered healthcare’ setting. The experience of insufficient access for children and young people was the result of adults not allowing access (whether knowingly or not). Parents, on the other hand, found their children to be non-participants due to professionals not optimizing the access to the care in relation to the family schedule or the child’s varying alertness throughout the day. While the parents often attributed non-access to professionals and the rehabilitation service system, the professionals experienced insufficient access to physically meet the child. This was sometimes due to the fact that parents did not see the point in child participation, but also since the rehabilitation service system demanded to see parents more than the child, or required resources that parents could not match (fitting activities into the family schedule or providing transports).

#### 3.1.1. Controlling Adults

The subcategory ‘Controlling adults’ emerged from the participants’ experiences of adults that in different ways did not share control or allowed the child to speak their mind. Even though the children expressed a desire for parents and professionals to support them in various situations in their rehabilitation, they found that adults sometimes overstepped the line between being supportive and being controlling. Children and young people expressed a dislike towards adult control, referring to the risk of adults assuming they know everything without asking the child:


*I think they almost spoke more to my parents than me actually […] I hate it when they’re talking with my parents over my head as though I’m not there.*
 (Young person No. 8)


*But the grownups! They don’t know what the child wants. Maybe they think they do, but they never actually do.*
 (Child No. 17)

A general concern for the professionals was about parents not allowing their children access to participation. This could be consciously by verbally expressing a desire to attend meetings without the child, or unconsciously by speaking on behalf of the child. Professionals as well as children and young people experienced parental presence as something that could quieten the child:


*Or that sometimes the child perhaps wants to say something that the child doesn’t want them (the parents) to hear.*
 (Professional No. 4)

Parents could also relate to having difficulties allowing the child to participate. While they expressed a cognitive understanding of children having to learn to take part in discussions involving themselves, there could be emotional barriers to letting go of advocating for their child or stopping being a helicopter parent. There could also be a wish to withhold information that motivated the parent to exclude their child: 


*Sometimes, in these situations, I don’t find it necessary for the child to attend. If they understand that the exercises, they are doing are aimed at improving their mobility, it becomes a task you have to do. But if you turn it into a game, and the child is unaware… it will just be fun, and the child doesn’t have to know.*
 (Parent No. 10)

Another aspect expressed by both professionals and parents was that of adults having a ‘medical veto’; when it was medically important, the child’s opinion or permission was not seen as vital. This veto could also overrule a child’s wish not to participate.


*And there are things that you can’t really change; that aren’t enjoyable. But maybe you can make the situation feel safer for the child.*
 (Professional No. 1)


*The children often feel that they are somewhat forced to go there. The children have to go there, they can’t choose because they’re small children.*
(Professional No. 4)

All of the participants expressed concerns regarding the risk of parents ‘taking over’ and controlling the situation if they are present in the room together with the child—thus indicating that it is sometimes better to leave the parent outside the meeting or examination room.

#### 3.1.2. Adult-Centered Healthcare

The subcategory ‘Adult-centered health care’ is derived from the experiences of the rehabilitation services being more suited to adult needs than child needs. This could be in relation to the environment (insufficient adaptations, non-stimulating conference rooms), timing (meetings were held at times that suited the adult instead of the child), or communication (the views of parents were asked for instead of those of the children). When children experienced adult-centered healthcare and were excluded from information-sharing and decision-making, it led to frustration and sometimes anger. The following child was asked how she/he feels when adults make decisions for them:


*It feels saddening… It doesn’t feel good [child gets emotional and sighs deeply].*
 (Child No.17)

Other children described the presence of tokenistic choices, such as being allowed to decide which cartoons to draw after an adult-centered meeting. The young people commented on the stressful environment at the rehabilitation services, where professionals did not have time to adapt the tempo of the meeting to suit the child: 


*They often say ‘well, we have so little time’. But I don’t think that I’m actually slower than the others—I might have bad hearing but I’m not slow. I think they should talk to me. I think this is why I don’t attend anymore [the rehabilitation services*
*].*
 (Young person No. 8)

The views shared by the children and young people were relatable for both parents and professionals, who could express frustration over the pediatric rehabilitation services being adapted mainly to adults’ needs and preferences:


*Another obstacle is that sometimes we only see the parents—and then it’s very difficult to enhance the child’s participation… And sometimes we try to get them to attend the meetings, but some children find it upsetting to be with so many adults… It could also be because the meetings are too long—the child can’t just sit still the amount of time that we adults want to spend talking.*
 (Professional No. 2)

### 3.2. Insufficient Trust

The category ‘Insufficient trust‘ includes both how children, young people, parents, and professionals distrust each other, and the low self-confidence experienced by children. Sometimes the children did not trust their own abilities, or adults did not trust the child to be able to participate. It could, however, also be that children did not trust adults to serve their best interests. The experience of having insufficient trust for the children, either from or towards adults, or experiencing low self-confidence, led to feelings of resignation towards participating in their rehabilitation.

#### 3.2.1. Low Level of Trust in Adults

Having a low level of trust in adults was expressed by young people, parents, and professionals. However, when the children talked about trusting adults it was often in positive words. The feeling of being excluded and viewed as an outsider was, however, for the young people a prominent experience from their time at the pediatric rehabilitation services. The young people could express how they were made to feel alienated and how adults seemed to pry into their personal space, resulting in a desire not to attend the pediatric rehabilitation services at all:


*Because you sit there, so many of them, and you’re supposed to talk about yourself. And they’re so curious! I didn’t like it. Having to talk again and again, and they still keep asking about you. So, no thanks, I’ll call when I need your help.*
 (Young person No. 2)

A concern among parents was sometimes raised regarding the competence of the professionals. Some parents had negative experiences from when professionals were not suitable to interact with their children. This could be due to a tendency to address the child in a demeaning manner, or simply lacking sufficient knowledge about how to communicate with children using alternative and augmentative communication, thus resulting in low a level of trust in the professionals:


*But after three [meetings] the psychologist thought that it was meaningless. So we discontinued the actual examination after the three meetings. Then we got in touch with our contact person and said that they were appalling. […] But you’re not to ignore the person [the child]*
*and assume that it’s easier to talk to me. That’s something that’s actually unacceptable.*
(Parent No.10)

Similarly, the professionals sometimes spoke of a low level of trust in the parents, questioning whether they wanted to include the child in healthcare decisions. 


*If child participation isn’t in their [the parents’] interest, and they instead feel that they are the ones in charge, then the child won’t be given the necessary space to be able to participate.*
 (Professional No. 3)

#### 3.2.2. Low Level of Trust in Children, and Low Self-Confidence in Children

The adults’ low level of trust in the children’s abilities to participate was brought up by children, young people, and adults alike. Some children considered themselves capable of participating, but they felt adults did not always see this competence. For other children, their own low self-confidence was a barrier. Some children wished that they were more comfortable in telling adults what they wanted, not seeming to reflect on the adults’ responsibility to empower them:


*I’ve mostly trusted the staff, and been like ‘oh well, they decide’, and such… I’d like to make decisions by myself, but I haven’t really dared to speak my mind.*
 (Child No. 13)

The young people could see adults’ low level of trust in children’s abilities as a negative influence on a child’s possibility to participate. However, they did not share the children’s experience of low self-confidence. Instead, they focused on how adults, both parents and professionals, need to trust children to be able to speak their mind. When they had chosen to withdraw, it was viewed as a result of having been excluded for too long and losing trust in the rehabilitation services:


*If the therapist tells you ‘This is what we’re going to do, you don’t get to decide’—then you’re being taught that you can’t affect change [in your*
*care].*
(Young person No. 1)

An explanation as to what hinders their child from participating in their rehabilitation was, according to the parents, the child’s own difficulties. The low level of trust in the child’s abilities made the parents take on more responsibilities themselves. The perception of the child lacking necessary skills could be due to the child’s cognition, young age, language skills, or dependence on adults who knew them to interpret their opinions. 


*Obstacles to participation… first and foremost, it’s that she can’t express herself, what she wants and doesn’t want… and then concentration. She becomes distant, loses focus and shuts down.*
 (Parent No. 12)

Some parents acknowledged their own lack of understanding of their child’s potential for participation as an obstacle:


*It’s very very difficult to know, how he functions and how he thinks [in order to be able to help the child to participate]. We still find it very hard to understand, despite him being as old as he is.*
 (Parent No. 9)

The professionals expressed similar concerns to those of the parents, with the main reason for children not participating in their rehabilitation being that they either lack the abilities to participate or that adults believe that they lack these abilities. Some professionals had doubts regarding the childrens’ ability to express views that were not influenced by adult opinions, and children were therefore assumed to have difficulties expressing their own views. The child’s ability to confidently express her/his own needs and to participate was also assumed to depend on the child’s developmental stage. 


*It’s completely based on the child’s age and cognitive skills. That’s what affects their level of participation.*
 (Professional No. 8)

### 3.3. Insufficient Involvement

The category ‘Insufficient involvement’ includes the subcategories ‘Norms of non-participation’, mostly presented by the children, and ‘Low level of commitment in children’ which was more prominent in the adults’ perspectives. For children, being insufficiently involved was attributed to feelings of indifference or insecurity towards participation. The children also expressed a desire to avoid responsibility, ‘boring’ meetings, and a tendency to conform to familiar routines and norms where adults set the agenda. All the participant groups believed that not understanding the purpose of one’s rehabilitation could lead to a low level of commitment. Adults and professionals also experienced that children sometimes lacked interest and motivation, and expressed resignation when children showed a low level of commitment.

#### 3.3.1. Norms of Non-Participation

The subcategory ‘norms of non-participation’ was mostly prominent in the children’s statements, where indifference towards participation was sometimes apparent. Children expressed ambivalence about the value of participating in decision-making and seemed to struggle with imagining why and how this could happen. The norm for these children was to not participate in decision-making, and it proved difficult to rethink this norm. Some children explained that their participation was not necessary and expressed trust in that adults would act in their best interest. When the professionals had already made decisions for them, it felt safe since they have more knowledge, and the children appreciated their work so far. Being told what to do could also feel less demanding for some children, who trusted adults to take care of ‘boring’ activities (talking, planning) on their behalf, so they did not have to bother.


*Oh, they told me what to do, about the training when we were training. They’ve told me, so I don’t need to decide.*
 (Child No. 10)


*Mm, children aren’t allowed to do that when you’re eight, aren’t allowed to make decisions, it’s only adults who do and make decisions.*
 (Child No. 12)


*I haven’t really dared to say what I think. I don’t really know, because … I’ve mostly trusted the leaders and said—well, they decide.*
(Child No. 13)

However, if the children should disagree with what had been decided, they believed they had a possibility to object and be given alternatives. Some young people were not as positive regarding the possibility to object to adults’ decisions, based on experiences from pediatric rehabilitation services where their opinions had been ignored. Young people also saw a risk of norms being reinforced unless they were allowed and trained to participate from an early age:


*I think it’s very important that you start when you’re young. And that the habilitation work with the children’s communication from the start and that … Perhaps if they ask the child during the meetings and say to the parents that you should be quiet now.*
 (Young person No.1)

#### 3.3.2. Low Level of Commitment in Children

The subcategory of ‘Low level of commitment in children’ was based on the participants’ perceptions that children had no interest or willingness to participate in rehabilitation-related decisions. The children accepted and could perform the exercises given to them but were regarded as having no interest in affecting their rehabilitation. The children often described their low level of commitment as being caused by experiences of ‘boring’ adult meetings. 


*For me it’s better to just be told what to do.*
 (Child No. 6)


*Ah, but it’s so boring [to attend meetings]!*
 (Child No. 13)


*No, boring [to participate in planning meetings].*
 (Child No. 20)


*I only want to decide about football.*
 (Child No. 5)

Some young people expressed similar thoughts, explaining that formal meetings were highly unmotivating for a child. Another explanation for experiencing a low level of commitment to their pediatric rehabilitation was that they had not really understood the purpose of their rehabilitation when they were children, and therefore did not find it important:


*Yes, I don’t know if I cared so much about it, I just did it, I don’t think I really cared about why then. […] Yes, of course, it’s always important to be allowed to participate and affect your care… but I think with many children, as with me, I just didn’t feel that the rehabilitation work was important. It was more or less the same, to me.*
 (Young person No. 8)

Parents and professionals shared the same experiences of children lacking interest or being unmotivated to attend training and meetings. This could be due to the child not being able to understand the purpose of participating, or genuinely not caring about their rehabilitation. Professionals were concerned that the children’s low level of commitment caused them to just follow their parents’ wishes:


*Throughout the years, I have met these children who, when you give them choices, only choose what they believe the parents want them to choose.*
 (Professional No. 6)

When children opted out of meetings, parents and professionals sometimes tried to gather their opinions beforehand. This proved to be difficult too when the children were perceived as not caring at all:


*For him, I actually don’t think he cares [about participating or not]… We usually try to get him to talk and express his feelings… but he doesn’t have many ideas… He simply doesn’t have anything to comment on.*
 (Parent No. 6)

## 4. Discussion

This study, aiming to explore barriers to child participation within pediatric rehabilitation through the perspectives of children, young people, parents, and professionals, found that there are insufficient prerequisites for child participation. Barriers to children’s participation were insufficient access, insufficient trust, and insufficient involvement. The results show that the participant groups were able to describe similar barriers but with different explanations as to where and how these barriers originate. Children spoke of adult-centeredness in healthcare, controlling adults, and norms of non-participation. Young people had similar experiences from pediatric rehabilitation but expressed less trust in professionals and the pediatric rehabilitation services in general. Parents and professionals shared a low level of trust in children and mainly attributed low participation to the children’s disabilities. In addition, parents expressed that children lacked commitment, but also had concerns regarding the adult-centeredness of the rehabilitation services and lack of skills in professionals. Professionals also found the rehabilitation services to be too adult-centered and inflexible but did not identify any professional shortcomings. However, as with children and young people, professionals highlighted controlling adults as an important barrier and related this to parents’ difficulties to share control.

The findings highlight the importance of supporting participation on multiple levels and in different situations in order to meet the requirements of the UNCRC. In line with previous research, children in this study consider themselves capable to be involved but are often excluded by norms of non-participation [16,33]. Parental presence was described as a barrier by all participant groups in this study, as found in the results from similar pediatric contexts [6]. The fact that children had more trust in adults than the young people suggests a gradual decline in trust over time for children with low access to participation, supporting findings that participation brings feelings of comfort, confidence and trust for children in healthcare situations [6,17,33]. For children in rehabilitation, however, a combination of deep trust towards adults and low self-confidence can be a considerable risk factor for choosing or accepting low participation, thus entering a life of exclusion already from a young age [34,35]. It thus becomes important to actively counteract this downward spiral. 

Changing norms takes time, but inclusive participatory processes can help build the confidence needed [8,35], and make children aware of their right to participate [36]. All of the participant perspectives in this study are important to address when developing supportive tools, methods or formats for future pediatric rehabilitation services. It is vital for generating change, however, that all actors believe in barriers being changeable, and participation malleable [7,22]. This study reveals a clear divergence between the perspectives of children and adults in terms of how context-bound barriers are. A medical model of disability [37,38] (predominantly expressed by adults) is thus still used to explain and reproduce norms of non-participation. 

Furthermore, this study shows that adult-centeredness in the rehabilitation services is a significant obstacle for child participation. This goes in line with a recent systematic review which shows that power imbalances, deficiencies in time, and professionals’ lack of shared decision-making strategies are key barriers when striving to implement a culture of increased participation in pediatrics [18]. The present study reveals that for children and young people with disabilities, their parents and professionals can identify similar barriers within pediatric rehabilitation. Both parents and professionals denounce the current system as unsuitable for promoting child participation and perceive it as static and beyond their control. At the same time, they conclude that some medical issues need to be discussed among adults in an efficient manner, indicating that child participation is not valid for all topics. Discussions about what participation in pediatric rehabilitation should contain is clearly needed [5]. The findings also highlight a need for discussions about social versus medical models [22] and indicates that a social model might be favorable when addressing barriers related to norms within the practice and prerequisites for child participation [5]. Future research may thus benefit from using this perspective. 

The divergence in perspectives and the expressed difficulty for adults to understand children’s abilities also calls for strategies, methods and tools that support interaction and communication between children and adults. The children in this study who were skeptical about participating in decision-making found themselves to be doing just that when given alternatives. However, alternatives must be fair and unbiased in order to shape one’s own opinion [18,35]. Aligned with principals in shared decision-making (SDM) tools [39], we suggest that support for participation should go beyond alternative and augmentative communication (AAC) and also facilitate a higher level of participation—in helping identifying goals, making choices, shaping opinions and setting agendas. Digital tools can bridge interaction barriers for people with disabilities [40,41] and might be superior to face-to-face interaction for children in healthcare contexts [5,16]. We thus suggest the development of digital support tools to address some of the barriers described in this paper, such as parents’ difficulties to share control, something which was expressed by all participant groups, indicating a need for new participatory formats. Support tools must however be flexible and aligned with the goals and preferences of the individual user [39,42], and user involvement in such developments is thus crucial.

### 4.1. Methodological Considerations

The insights drawn from multiple stakeholders’ perspectives can guide efforts to mitigate the identified barriers and help avoid reinforcing norms of non-participation. The findings are also of value for designers and researchers developing alternative participatory formats within pediatric rehabilitation. For example, they can inform participatory design methods, the design of meeting settings or (digital) support tools aiming to increase child participation in rehabilitation. Further research related to development in these areas is encouraged.

The fact that our sample came from rehabilitation services in just one region might have had an impact on the generalizability of our results, in the sense that there could be differences in working practices or approaches between regions. A limitation of this study is also the variety in the amount of data obtained from different participants. Parents and professionals contributed with more input than children and young people. There were also differences within the young participants where some simply did not speak much (including use of AAC). This is not a surprise, considering that many children found long conversations with adults as ‘boring’. Individual interviews proved favorable to focus groups, since it was technically challenging to get a discussion going in groups where a variety of AAC was used. The interview formats used in this study thus contributed to this limitation and further highlights the need for flexible methods when involving children and young people with disabilities [41].

### 4.2. Implications

The results of this study may help rehabilitation professionals and researchers to understand what influences children’s possibilities to participate. Many of the identified barriers are naturally related to each other, and overcoming one barrier might generate positive effects elsewhere. Possibly, most urgent to overcome are barriers created by a culture of adult-centeredness and the problem of controlling adults. Intertwined in this task is to increase trust in children, since sharing control is difficult if this trust is lacking. These findings imply that a child-centered approach should be at the core of pediatric rehabilitation. 

## 5. Conclusions

Through the perspectives of children, young people, parents and professionals, this qualitative study identifies insufficient prerequisites for child participation in pediatric rehabilitation. Key barriers to child participation are insufficient access, insufficient trust, and insufficient involvement. The experiences shared by children and young people with disabilities, their parents and healthcare professionals also show divergences in perspectives between participant groups in terms of where and how barriers originate and for which subjects child participation is appropriate. Adult-centeredness in the rehabilitation services and parental presence were described as barriers by all participant groups, indicating a need for new participatory formats. When addressing this we see a potential in digital support tools tailored to the preferences of children with disabilities, and we welcome further research in this field.

## Figures and Tables

**Table 1 ijerph-18-10119-t001:** Demographic and sociodemographic data of participants.

Variables	Children (*n* = 20)	Young People (*n* = 8)	Parents (*n* = 17)	Healthcare Professionals (*n* = 9)
**Sex** (*n*)				
Female	9	6	13	9
Male	11	2	4	0
**Age years Median** (Range)	12 (6–16)	24 (19–30)	43 (31–62)	
**Main disability** (*n*)			(Child’s main disability)	
Physical disability	8	5	6	
Intellectual disability	7	0	7	
Autism spectrum disorder	5	3	4	
**Experience of pediatric rehabilitation** (years)				
Median (Range)	6.5 (0.1–16)		5.5 (0.1–16)	
**Native-born** (*n*)	17	8	15	
**Foreign-born** (*n*)	3	0	2	
**Civil Status** (*n*)				
Co-habiting			12	
Living alone			5	
**Education Level** (*n*)				
Lower Secondary School			0	0
Upper Secondary School			8	0
University/University College			9	9
**Employment** (*n*)				
Full time			7	
Part-time			9	
Parental leave			1	
**Age of child in pediatric rehabilitation** (years)				
Median (Range)			13 (6–16)	
**Sex of the child** (*n*)				
Female			8	
Male			9	
**Profession** (*n*)				
Special Education				1
Occupational Therapist				1
Registered Nurse				1
Psychologist				2
Physiotherapist				2
Speech Therapy				1
Social worker				1

**Table 2 ijerph-18-10119-t002:** Barriers to participation according to children and young people with disabilities, parents, and professionals.

Theme	Insufficient Prerequisites for Child Participation		
**Categories**	Insufficient access	Insufficient trust	Insufficient involvement
**Subcategories**	Controlling adults	Low level of trust in adults	Norms of non-participation
		Low level of trust in children	
	Adult-centered healthcare	Low self-confidence in children	Low level of commitment in children

**Table 3 ijerph-18-10119-t003:** Dominant subcategories, by participant group.

Participant Group	Dominant Subcategories
Children	Adult-centered healthcare, Controlling adults, Norms of non-participation
Young people	Adult-centered healthcare, Controlling adults, Low level of trust in adults
Parents	Low level of trust in children, Low level of commitment in children, Adult-centered healthcare
Healthcare professionals	Low level of trust in children, Adult-centered healthcare, Controlling adults

## Data Availability

Not applicable. The data will not be shared as ethics approval for the study requires that data files and the transcribed interviews are kept in locked files, accessible only to the researchers.

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
