# Peer review of "Exploring Barriers to Participation in Pediatric Rehabilitation: Voices of Children and Young People with Disabilities, Parents, and Professionals"

_ijerph, 2021, doi:10.3390/ijerph181910119_

Round 1

Reviewer 1 Report

I am happy to have been provided the opportunity to read and review this manuscript.

Thank you for submitting your manuscript “Barriers to participation as told by children and young people with disabilities, their parents, and professionals: a qualitative study.”  This manuscript highlights the importance of ensuring that all students with disabilities, have access to Pediatric rehabilitation.

The authors are commended for the following aspects of their manuscript:

Overall, procedures and data analysis were well described but a bit disorganized.

Extracts used throughout are excellent examples that highlight some of the key themes identify by the researchers.

The authors should consider general proofreading for punctuation and structure throughout the paper.

The authors are encouraged to review the structure of the method section under the headings I have provided.

My view is that this article has potential for publication. The following areas need to be revised and expanded upon by the authors before publication.

Topic

I think the topic should be: “Exploring the Barriers to participation in Pediatric rehabilitation: Voices of children, young people with disabilities, parents, and professionals”.

Abstract

Line 9 – missing comma ‘rehabilitation, it is”

Line 11 – change ‘within’ to ‘in’

Line 13 – can you please give separate numbers for those who took part in ‘individual interview (n=__) and focus groups (n=_).

Line 14 – are you sure you used content analysis? It looks like you actually used thematic analysis.

There is lack of information of the conceptual framework you are using a lens to identify the barriers to participation – the social model of disability must be described in the abstract as the theoretical lens.

Also there is no information in the abstract as to how the participants were recruited for the interviews and focus group discussion – did you use ‘purposive, convenient, snowball sampling approach?

Introduction

Line 52 to 53 – the sentence “in that these might be perceived as fixed [22].” Does not seem to be complete. Please replace the word ‘fixed’ with ‘deficit view”.

Please provide a section on your conceptual framework – social model of disability.  You have to explain this concept and do some literature review on how this concept have been used to identify barriers to inclusion for persons with disability and helped improve participation. You need to provide some justification for its use.

  1. Materials and Methods

2.1. Design and Participants

Please explain which participant you conducted interviews with and how many and those you conducted focus group discussion with.

Please it will be nice to make some reorganization on this section: please the ideas are a bit disorganized and I would like to see them organize under the following headings:

  • Study participants
  • Study design
  • Instruments – please describe the instruments for the data collection and how you developed them
  • Data collection – please describe how were the participants recruited and data collection procedure
  • Data analysis

Data analysis

You data analysis approach sounds more like thematic analysis.

Results

The result section is well-presented.

Discussion

Please remove “4.1 Main results”.

Can you please provide just a brief background to the theoretical framework? Before moving on to discuss the barriers.

Limitations

Please could described some of the limitations in the study – issues regarding the research design, sampling, and what future research should focus on – I can see the last two sentences under the “Implications” relate to this.

Implications

Please I would like to see some description of policy implications of these findings.

Conclusions

Well-written.

Author Response

Response to Reviewer I Comments

I am happy to have been provided the opportunity to read and review this manuscript.

Thank you for submitting your manuscript “Barriers to participation as told by children and young people with disabilities, their parents, and professionals: a qualitative study.”  This manuscript highlights the importance of ensuring that all students with disabilities, have access to Pediatric rehabilitation.

The authors are commended for the following aspects of their manuscript:

Overall, procedures and data analysis were well described but a bit disorganized.

Extracts used throughout are excellent examples that highlight some of the key themes identify by the researchers.

The authors should consider general proofreading for punctuation and structure throughout the paper.

The authors are encouraged to review the structure of the method section under the headings I have provided.

My view is that this article has potential for publication. The following areas need to be revised and expanded upon by the authors before publication.

Topic

Point 1. I think the topic should be: “Exploring the Barriers to participation in Pediatric rehabilitation: Voices of children, young people with disabilities, parents, and professionals”.

Response 1: We thank the reviewer for this suggestion and have used this title.

Abstract

Point 2. Line 9 – missing comma ‘rehabilitation, it is”. Line 11 – change ‘within’ to ‘in’. Line 13 – can you please give separate numbers for those who took part in ‘individual interview (n=_) and focus groups (n=_).

Response 2: We are grateful for both the corrections and suggestion, and has changed the abstract accordingly.

Point 3. Line 14 – are you sure you used content analysis? It looks like you actually used thematic analysis.

Response 3: We agree that qualitative content analysis is similar to thematic analysis. We have, however, followed the analysis steps of Graneheim et al. (2017) and Graneheim and Lundman (2004) which we consider to be an established methodological approach for qualitative content analysis.

Point 4. There is lack of information of the conceptual framework you are using a lens to identify the barriers to participation – the social model of disability must be described in the abstract as the theoretical lens.

Response 4: Even though we introduce the social model in the introduction, this is to create an understanding of the area (barriers in relation to disabilities) and to offer examples of perspectives within this discourse. We did not use a particular theoretical framework in the analysis, but rather an inductive qualitative content analysis (see 2.1 and 2.4). However, since this appear to be unclear to the reader, we have changed the sentences at line 56-61 in the introduction to:

“Critical perspectives such as a social model of disability redirects the focus towards social environments. In this perspective, barriers are not an effect of an individual’s disability but by societal exclusion or contextual norms. This model highlights the importance of empowering support.”

In addition, this could be an idea for forthcoming studies, i.e. to have an abductive approach for analysis and use a social model framework to support it. This suggestion has subsequently been added to the discussion at lines 500-503 (see response 13).

Point 5. Also there is no information in the abstract as to how the participants were recruited for the interviews and focus group discussion – did you use ‘purposive, convenient, snowball sampling approach?

Response 5: Thanks for highlighting this. Since we have reached the word limit in the abstract, we have added this information in 2.2 Participants, line 78-79: “A purposeful sample of 54 participants were included […]”.

Introduction

Point 6. Line 52 to 53 – the sentence “in that these might be perceived as fixed [22].” Does not seem to be complete. Please replace the word ‘fixed’ with ‘deficit view”.

Response 6: The sentence in question has been clarified, line 51-53: “[…] in that these barriers might be perceived as static”.

Point 7. Please provide a section on your conceptual framework – social model of disability.  You have to explain this concept and do some literature review on how this concept have been used to identify barriers to inclusion for persons with disability and helped improve participation. You need to provide some justification for its use.

Response 7: Please see response 4 regarding the theoretical framework.

Materials and Methods

2.1. Design and Participants

Point 8. Please explain which participant you conducted interviews with and how many and those you conducted focus group discussion with.

Response 8: Data was collected in individual interviews(n=48) and focus groups (n=8), which has been clarified in the abstract (line 13-14) and in 2.3 Data Collection (line 98-99).

Point 9. Please it will be nice to make some reorganization on this section: please the ideas are a bit disorganized and I would like to see them organize under the following headings:

  • Study participants
  • Study design
  • Instruments – please describe the instruments for the data collection and how you developed them
  • Data collection – please describe how were the participants recruited and data collection procedure
  • Data analysis

Response 9: We have tried to align our structure to the style of the journal. In principle we follow the organization suggested here, albeit we start with Study Design. And we now separated “Design and Participants” into “2.1. Study Design” and “2.2. Participants”. “Instruments” might in this qualitative context be equivalent to our interview guide, described under 2.3. Data Collection (line 99-104).

Data analysis

Point 10. You data analysis approach sounds more like thematic analysis.

Response 10: We agree that qualitative content analysis is similar to thematic analysis. We have, however, followed the analysis steps of Graneheim et al. (2017) and Graneheim and Lundman (2004) which we consider to be an established methodological approach for qualitative content analysis.

Point 11. Results

The result section is well-presented.

Response 11: Thank you!

Discussion

Point 12. Please remove “4.1 Main results”.

Response 12: We have removed the subheading 4.1. (line 450).

Point 13. Can you please provide just a brief background to the theoretical framework? Before moving on to discuss the barriers.

Response 13: As mentioned in response 4, we did not use a particular theoretical framework to analyze the data. But since the results goes well in line with a social model, and it could be an idea for forthcoming studies to use this perspective, the following sentence was edited at lines 500-503:

“The findings also highlight a need for discussions about social versus medical models [22] and indicates that a social model might be favorable when addressing barriers related to norms within the practice and prerequisites for child participation [5]. Future research may thus benefit from using this perspective.”

Limitations

Point 14. Please could described some of the limitations in the study – issues regarding the research design, sampling, and what future research should focus on – I can see the last two sentences under the “Implications” relate to this.

Response 14: We have inserted the subheading “Methodological Considerations” and moved the last section of the Implications to this section (now line 521-527). As suggested, we also added a sentence about limitations related to the sampling (line 528-530): “The fact that our sample came from rehabilitation services in just one region might have had an impact on the generalizability of our results, in the sense that there could be differences in working practices or approaches between regions.”

Implications

Point 15. Please I would like to see some description of policy implications of these findings.

Response 15: We have added the following sentence at the end of the Implications (line 547-548): These findings imply that a child-centered approach should be at the core of pediatric rehabilitation.”

Conclusions

Point 16. Well-written.

Response 16: We appreciate this.

Reviewer 2 Report

The paper was a well-constructed and conducted qualitative study on barriers to physical activity for children in a rehab setting. I particularly liked the fact that the authors interviewed, parents, children with disabilities and health care workers.

-Large sample size, followed best practice in qualitative research. Results were clearly presented with lots of quotes from participants.

-Well written with excellent English skills. References were appropriate for this paper. I did not see any places for improvements. The title could have specific physical activity rehabilitation.

-The paper is well written, and the study was nicely conducted and presented.

My only suggestions for changes is to the title. The title needs to add barriers to participation in "rehabilitation." 

Author Response

Response to Reviewer II Comments

The paper was a well-constructed and conducted qualitative study on barriers to physical activity for children in a rehab setting. I particularly liked the fact that the authors interviewed, parents, children with disabilities and health care workers.

-Large sample size, followed best practice in qualitative research. Results were clearly presented with lots of quotes from participants.

-Well written with excellent English skills. References were appropriate for this paper. I did not see any places for improvements. The title could have specific physical activity rehabilitation.

-The paper is well written, and the study was nicely conducted and presented.

Point 1. My only suggestions for changes is to the title. The title needs to add barriers to participation in "rehabilitation." 

Response 1: We are grateful for this commendation. We also appreciate the suggestion regarding the title, which has been changed to include the word “rehabilitation": Exploring barriers to participation in pediatric rehabilitation: Voices of children, young people with disabilities, parents, and professionals.

Reviewer 3 Report

General comments

The study is an interesting contribution to the discourse on  the barriers  to participation within paediatric rehabilitation services. The  use of multiple stakeholder highlights the differences in opinions and the need for further discourse on how to promote real participation for children and young persons with disabilities.

I have some specific comments and recommendations:

Specific Comments

Overall, the manuscript is well written, however, there are many typographical and grammatical errors. I have highlighted a few and recommend that the author(s) critically review the manuscript for errors.

barriers for participation(line 11) should be barriers to

The child and its disability(line 51)   

change to her/his as used in  child with her/his unique (line 47)

Method

explorative, inductive design????(Line 73)

This is unclear and does not  reflect what is described in the method section. I recommend the authors change this to explorative qualitative study

The focus groups included two and five young people respectively.(line 110-111)

What does respectively refer to?

Tables 1-3 can be combined into one; it will make it easier to read and compare the demographics

The title of table 4(line 160-162) should be shortened to a descriptor that concisely  describes the table

child was asked how it feels when(line 238) ??? changer it to his or her and throughout the manuscript.

Results

The study results are interesting and it would have been more interesting to use a framework to situate the findings and the differences in opinion between the multiple stakeholders. How different were the findings from the individual interviews from the FGDs? Was an attempt made to discuss and explore the  differences in opinions by the different stakeholders?

Discussion

Furthermore, this study shows that adult-centredness in the rehabilitation services is  strongly connected to a low level of commitment in the children. A recent systematic re- view shows that power imbalances, deficiencies in time and professionals’ lack of shared decision-making strategies are key barriers when striving to implement a culture of increased participation in pediatrics (lines 487-491)

It is hard to understand the relationship between the first sentence (study finding) and the cited systematic review. Perhaps the authors, can clearly state what – low level of commitment in the children means in the sentence.

The proposed use of digital support tools to bridge barriers in interaction is interesting. It will be fascinating to see  how it is adopted by children/young people and their parents and the impact of digital divide on its usefulness.

Author Response

Response to Reviewer III Comments

General comments

The study is an interesting contribution to the discourse on  the barriers  to participation within paediatric rehabilitation services. The  use of multiple stakeholder highlights the differences in opinions and the need for further discourse on how to promote real participation for children and young persons with disabilities.

I have some specific comments and recommendations:

Specific Comments

Point 1. Overall, the manuscript is well written, however, there are many typographical and grammatical errors. I have highlighted a few and recommend that the author(s) critically review the manuscript for errors.

barriers for participation(line 11) should be barriers to

The child and its disability(line 51)   

change to her/his as used in  child with her/his unique (line 47)

Response 1: We appreciate the highlighting of these errors and have corrected these throughout the manuscript. The manuscript was reviewed by a native English speaking reviewer.

Method

Point 2. explorative, inductive design????(Line 73)

This is unclear and does not  reflect what is described in the method section. I recommend the authors change this to explorative qualitative study

Response 2: The sentence was changed into “This study had an explorative, qualitative design.” (line 74).

Point 3. The focus groups included two and five young people respectively.(line 110-111)

What does respectively refer to?

Response 3: The sentence has been changed into ”One group had two participants and the other had six participants.” (line 114-115).

Point 4. Tables 1-3 can be combined into one; it will make it easier to read and compare the demographics

Response 4: We appreciate this suggestion and have merged the three tables into a single table.

Point 5. The title of table 4(line 160-162) should be shortened to a descriptor that concisely  describes the table

Response 5: The table caption (now Table 2, line 164-165), has been changed into “Barriers to participation according to children and young people with disabilities, parents, and professionals.”

Point 6. child was asked how it feels when(line 238) ??? changer it to his or her and throughout the manuscript.

Response 6: We have removed “how it feels when” to avoid this misperception. The sentence was changed to  ”The following child was asked how she/he feels when adults make decisions for them” (line 240-241).

Results

Point 7. The study results are interesting and it would have been more interesting to use a framework to situate the findings and the differences in opinion between the multiple stakeholders. How different were the findings from the individual interviews from the FGDs? Was an attempt made to discuss and explore the  differences in opinions by the different stakeholders?

Response 7: We believe that the differences in opinion is one of the main findings of the study. In the analysis and table 3 (previously table 5) we try to explore and categorize these differences. While it was not within our scope to explain them further, we do believe that using for example a social model perspective could be fruitful for future research, especially from what the young participants discussed. Regarding the two formats – focus group and individual interview – we have added a sentence in the Discussion, 4.1. Methodological Considerations (line 535-537): “Individual interviews proved favorable to focus groups, since it was technically challenging to get a discussion going in groups where a variety of AAC was used.”   

Discussion

Furthermore, this study shows that adult-centredness in the rehabilitation services is  strongly connected to a low level of commitment in the children. A recent systematic re- view shows that power imbalances, deficiencies in time and professionals’ lack of shared decision-making strategies are key barriers when striving to implement a culture of increased participation in pediatrics (lines 487-491)

Point 8. It is hard to understand the relationship between the first sentence (study finding) and the cited systematic review. Perhaps the authors, can clearly state what – low level of commitment in the children means in the sentence.

Response 8: The connection has been made clearer by replacing the first part (line 489-493): Furthermore, this study shows that adult-centeredness in the rehabilitation services is a significant obstacle for child participation. This goes in line with a recent systematic review which shows that power imbalances, deficiencies in time and professionals’ lack of shared decision-making strategies are key barriers when striving to implement a culture of increased participation in pediatrics [18].”

Point 9. The proposed use of digital support tools to bridge barriers in interaction is interesting. It will be fascinating to see  how it is adopted by children/young people and their parents and the impact of digital divide on its usefulness.s

Response 9: Thank you, we agree!